# The Influence of Long-Term Storage on the Epiphytic Microbiome of Postharvest Apples and on *Penicillium expansum* Occurrence and Patulin Accumulation

**DOI:** 10.3390/toxins16020102

**Published:** 2024-02-12

**Authors:** Reem Al Riachy, Caroline Strub, Noël Durand, Vincent Chochois, Félicie Lopez-Lauri, Angélique Fontana, Sabine Schorr-Galindo

**Affiliations:** 1Qualisud, Univ Montpellier, Univ d’Avignon, CIRAD, Institut Agro, IRD, Univ de La Réunion, Montpellier, France; riachy.reem@gmail.com (R.A.R.); caroline.strub@umontpellier.fr (C.S.); noel.durand@cirad.fr (N.D.); vincent.chochois@cirad.fr (V.C.); felicie.lauri@univ-avignon.fr (F.L.-L.); angelique.fontana@umontpellier.fr (A.F.); 2CIRAD, UMR Qualisud, F-34398 Montpellier, France

**Keywords:** apple, patulin, *Penicillium expansum*, epiphytic microbiome

## Abstract

Patulin is a secondary metabolite primarily synthesized by the fungus *Penicillium expansum*, which is responsible for blue mold disease on apples. The latter are highly susceptible to fungal infection in the postharvest stages. Apples destined to produce compotes are processed throughout the year, which implies that long periods of storage are required under controlled atmospheres. *P. expansum* is capable of infecting apples throughout the whole process, and patulin can be detected in the end-product. In the present study, 455 apples (organically and conventionally grown), destined to produce compotes, of the variety “Golden Delicious” were sampled at multiple postharvest steps. The apple samples were analyzed for their patulin content and *P. expansum* was quantified using real-time PCR. The patulin results showed no significant differences between the two cultivation techniques; however, two critical control points were identified: the long-term storage and the deck storage of apples at ambient temperature before transport. Additionally, alterations in the epiphytic microbiota of both fungi and bacteria throughout various steps were investigated through the application of a metabarcoding approach. The alpha and beta diversity analysis highlighted the effect of long-term storage, causing an increase in the bacterial and fungal diversity on apples, and showed significant differences in the microbial communities during the different postharvest steps. The different network analyses demonstrated intra-species relationships. Multiple pairs of fungal and bacterial competitive relationships were observed. Positive interactions were also observed between *P. expansum* and multiple fungal and bacterial species. These network analyses provide a basis for further fungal and bacterial interaction analyses for fruit disease biocontrol.

## 1. Introduction

Domesticated apples (*Malus x domestica*) are globally regarded as an important crop and constitute the largest fruit crop produced in temperate regions. In the last 20 years, the global production of apple has increased from 41 million tons in 1990 to 93.14 million tons in 2021 [1,2]. The most significant aspect of the apple is its ability to be stored for a long period of time, ensuring a sufficient supply of fresh fruit all year [3,4]. The average storage period for apples is usually a few months but can increase up to one year in cold conditions and controlled-atmosphere regimes. However, long storage periods can lead to the infection of apples by pathogens, resulting in the development of decay [5]. Moreover, the nutritional composition of apples is favorable to the growth of pathogenic fungi, especially *Penicillium expansum*, which causes blue mold disease [6,7].

*P. expansum* is a necrotrophic fungus that will attack apples only after harvest, causing huge economic losses. It is ubiquitous in the soil and air and can be found on inert equipment in contact with apples [6]. Thus, the contamination of apples by this pathogen is mostly due to the spores found in the soil and debris on the fruit bins and equipment, as well as spores transmitted via water handling systems. The prevention of the propagation and development of *P. expansum* during storage is a major challenge in maintaining the quality and safety of the fruit. The main issue regarding this fungus is its capacity to produce patulin, a harmful mycotoxin to humans [8,9]. Patulin is produced during infection in the necrosis area but can migrate, and its contamination has been widely reported in both infected and uninfected areas of apples [10].

Patulin is immunotoxic and hepatotoxic and causes damage to the gastrointestinal tract and neurological problems [11,12]. The International Agency for Research on Cancer (IARC) classified patulin as group 3, “not classifiable as a carcinogen to humans” [13]. The European Union (EU) has set maximum levels of 50, 25 and 10 µg patulin/kg, respectively, for fruit juices, nectars and fermented apple beverages; solid apple products; and apple-based products for infants and young children [14]. Other countries and organizations have also set maximum permitted levels of patulin in food order to control the health risk. In the United States, patulin levels in pure and blended apple juice are limited to 50 μg/L by the Food and Drug Administration (FDA) [15]. Additionally, the Ministry of Health of the People’s Republic of China has set the maximum level for patulin in apple products at 50 μg/kg [16]. Moreover, a tolerable daily intake of 0.4 mg/kg body weight/day for PAT has been established globally by the Joint FAO/WHO Expert Committee on Food Additives [17]. Patulin is stable under acidic conditions and resistant to pasteurization; therefore, it is difficult to eliminate it using conventional processing in apple compotes and other products, resulting in a potential food safety hazard [18]. The incidence of patulin contamination remains high, despite global efforts to reduce the levels of this mycotoxin at each stage of the fruit production process. The factors involved in patulin production are diverse and complicated.

The study of the changes in the composition and assembly of the fruit microbiome associated with apples, during storage and postharvest treatment in stations and packinghouses, is important to better understand and control postharvest diseases. This would reduce the losses and wastage throughout the supply chain [19]. In fact, the apple microbiome is impacted by many factors, including abiotic conditions such as the climate, the soil type and nutrient availability, and management practices such as the type of harvesting, the sanitary conditions postharvest and the time and temperature during storage. The study of the fruit microbiome has become an important factor for breeding strategies and seed production and for the management and control of pre- and postharvest pathogens [20,21,22]. In the last few years, there has been growing interest in exploring the impact of microbial communities on both pre- and postharvest fruit quality, as well as its susceptibility or resistance to postharvest diseases [23]. In fact, changes in the structure of these microbial communities could be responsible for modulating the pathogenicity and virulence factors within the pathogen and suppressing the activation of the resistance responses in the host [24]. Therefore, the study of the epiphytic microbiome of apples at different postharvest stages will allow the assessment of the influence of the postharvest processing techniques on the apple microbiome, resulting in the development of new strategies to improve food safety and control major postharvest diseases responsible for economic losses.

In this study, apples of the variety “Golden Delicious”, destined to produce compotes, were followed up from the step after harvesting until the loading of the bin in the packinghouse prior to transportation. The aim of this work was to identify the means of contamination of apples by *P. expansum* in the postharvest stages and to evaluate the risk of patulin accumulation in the apple-based products. This study also aimed to evaluate the effect of long-term storage and multiple processing steps on the epiphytic microbiome of the apple and the effect of the latter on the growth and toxinogenesis of *P. expansum*. The impact of the management practices (organic and conventional) was also assessed. 

## 2. Results

### 2.1. Patulin Content Analysis on Sampled Apples by HPLC-UV

The different apples sampled after harvesting and after 6 months of cold storage at different postharvest steps were analyzed for their patulin content. For conventional apples, the number of positive samples was rather consistent throughout the entire production chain. Table 1 shows the results of the patulin analysis carried out on all apples at the different sampling stages. Two of the nine samples analyzed after harvesting contained patulin with values greater than the regulatory limit, and, at the opening of the cold chambers, the same proportion of apples was contaminated. The processing steps in the packinghouse, especially the use of multiple baths, seem to reduce the incidence of patulin. The latter increases after the transition to room temperature (deck storage), with two out of three positive samples exceeding the regulatory limit. 

For organic apples, no samples were found to be contaminated with patulin directly after harvesting. After 6 months of cold storage, three out of the nine analyzed samples showed patulin, with values exceeding the regulatory limit by up to 14 times in one of the samples. The processing steps seem to decrease the percentage of contaminated apples. However, one sample showed a significant amount of patulin after passing through the packaging line (189.2 µg kg^−1^). Deck storage seems to increase the percentage of contamination. Overall, in both organic and conventional apples, two steps seem to have a negative effect on the incidence of patulin: long-term cold storage under CA and the deck storage of apples. However, the Kruskal–Wallis test revealed no significant differences between the different steps and the different cultivation systems (Kruskal–Wallis/FDR, *p* > 0.05).

No patulin was found in the water samples of the different baths. However, a water sample was taken from the bin after apple loading, and the threefold analysis on it indicated a very high amount of patulin at the level of 221 ± 11 µg kg^−1^. This water sample was a result of condensation, when the apples transitioned from 4 °C to an ambient temperature of 25 ± 5 °C.

### 2.2. P. expansum Quantification on the Surfaces of Apples

The quantities of *P. expansum* DNA per g of apple were assessed on the surfaces of all samples using q-PCR (Figure 1). The number of positive samples and the values of the log copies of *P. expansum* DNA per g of apple in the different samples are presented in Table 2. For conventional apples, the number of positive samples exhibiting *P. expansum* on their surface demonstrated a relatively uniform pattern during both the harvesting and postharvest stages, with, on average, five positive samples out of nine. The *P. expansum* abundances recorded after harvest increased in the last two steps, with a mean value of 2.6 ± 0.2 log copies of *P. expansum* DNA/g of apple after harvest and 4.6 ± 1.8 log copies of *P. expansum* DNA/g of apple at the last step, after a period of deck storage. The transfer of apple samples through the various processing lines did not seem to influence the incidence of *P. expansum*. The results of the one-way ANOVA test indicated no statistically significant variances among the mean values of *P. expansum* abundance observed at each step (*p* > 0.05). 

The incidence of contamination after harvest and at different postharvest stages is different for organic apples compared to conventional apples, showing more significant changes for organic apples. After harvest, *P. expansum* was quantified on the surface of only one sample out of nine for organic apples, increasing after 6 months of storage, while five out of nine samples exhibited *P. expansum* on their surfaces, with a mean value five times higher than the value recorded after harvest. This increase was statistically significant (Tukey HSD test, *p* < 0.001). The transfer of apples through the various processing lines does not influence the incidence of *P. expansum.* However, the apples sampled after a period of deck storage exhibited the highest amounts of *P. expansum* DNA on their surfaces, at the level of 5.2 ± 1.3 log copies of *P. expansum* DNA/g of apple. At this stage, also, 100% of the tested samples were positive for *P. expansum*. This increase was statistically significant (Tukey HSD test, *p* < 0.001). Therefore, two steps significantly increased the amount of *P. expansum* DNA on the surfaces of organically grown apples: the period of cold storage under CA for 6 months and the transfer of apples from 4 °C to an ambient temperature of 25 ± 5 °C (deck storage). 

There were no significant differences in the values of *P. expansum* DNA obtained in all organic samples compared to the conventional samples (one-way ANOVA, *p* > 0.05). It is important to emphasize that the quantification of *P. expansum* occurred in samples where decay was not visually apparent, and the apples exhibited an intact appearance.

### 2.3. Overall Composition of the Bacterial and Fungal Microbiota on the Surfaces of Apples

#### 2.3.1. Sequencing Results

After conducting paired-end alignments, implementing quality filtering, and eliminating chimeras, singletons, and mitochondrial and chloroplast sequences, a total of 395,010 16S rRNA reads for bacteria and 730,146 reads for fungal internal transcribed spacers (ITS) were successfully recovered. These reads were then assigned to 975 bacterial and 1214 fungal taxa amplicon sequence variants (ASVs), across a dataset comprising 118 samples. To standardize the dataset and eliminate sample heterogeneity, rarefaction was applied to achieve an even depth of 10,000 reads for ITS and 5000 reads for 16S per sample. Consequently, samples with fewer than 10,000 reads for ITS and 5000 reads for 16S were excluded from the respective analyses. The current sequencing depth proved adequate for both fungal and bacterial diversity analyses, as evidenced by the rarefaction curves approaching saturation in both cases. It is essential to note that the identified ASVs in this study were associated with the washed-off epiphytic community and not the entire host microbiome.

#### 2.3.2. General Overview of the Epiphytic Fungal and Bacterial Microbiota in All Apple Samples

A total of 157 fungal genera belonging to 3 phyla, 15 classes, 46 orders and 95 families were detected on the surfaces of all apples. Members of Ascomycota were the dominant fungal phyla across all samples, accounting for 85.69% of the total number of detected sequences. This was followed by *Basidiomycota* (14.3%) and *Mortierellomycota* (0.006%). All ASVs were predominantly identified as members of the classes *Dothideomyctes* (50.1%), *Leotiomycetes* (15.7%) and *Sordariomycetes* (11.1%), followed by *Tremellomycetes* (8.1%) and *Saccharomycetes* (3.5%). Moreover, *Cystobasidiomycetes* and *Eurotiomycetes* were detected at low abundances (2.5% and 2.2%, respectively). The most abundant genus was *Cladiosporium* (14.21%), followed by *Aureobasidium* (14.14%). The genus *Penicillium* occupied 11th position as the most abundant genus across all samples, with a percentage of 1.93%.

Furthermore, a collective count of 138 bacterial genera, distributed across 12 phyla, 25 classes, 28 orders and 68 families, was observed on the surfaces of all apple samples (both cultivation systems and at all the sampling steps). Bacterial ASVs, across all samples, were assigned to Pseudomonadota (82.89%), Actinomycetota (15.09%), Bacteroidota (1.39%) and Bacillota (0.39%). Other phyla were detected but at low frequencies, such as Verrucomicrobia, Cyanobacteria, Acidobacteria and others. The class Gammaproteobacteria (48.81%) was the most abundant across all samples, followed by Alphaproteobacteria (31.73%) and Actinobacteria (15.09%). The family *Pseudomonadaceae* was the most abundant across samples (30.01%), followed by *Enterobacteriaceae* (16.67%) (mainly at the first step, immediately after harvesting), *Sphingomonadaceae* (13.14%), *Microbacteriaceae* (12.05%) and *Acetobacteracea* (11.33%). The latter were mainly absent after harvesting and became dominant after 6 months of cold storage. Species belonging to the genera *Pantoea* and *Pseudomonas* were mainly abundant at the beginning of storage and significantly decreased after 6 months of storage, with the emergence of *Clavibacter* and *Sphingomonas*; after a period of deck storage, the genus *Gluconobacter* became dominant.

#### 2.3.3. Core Microbiome of Golden Delicious Apple

The core microbiome can be defined as all taxa present in at least 75% of all samples [19]. In this study, the core microbiome of the Golden Delicious apples across all samples consisted of seven fungal genera and seven bacterial genera. The fungal genera were *Aureobasidium*, *Vishniacozyma*, *Cladosporium*, *Leptosphaeria*, *Alternaria*, *Cadophora* and a non-identified genus. The bacterial genera consisted of *Pseudomonas*, *Sphingomonas*, *Clavibacter*, *Methylobacterium*, *Pantoea*, *Rhizobium* and a non-identified genus. The fungal genus *Aureobasidium* was prevalent in 100% of tested samples, followed by *Vishniacozyma*, found on the surfaces of 97.3% of all samples, and *Cladosporium* in 95.9% of the samples. Other genera were found at high percentages (but less than 75%), including *Cystobasidium* (74%), *Sporobolomyces* (71.23%) and *Paraphaeosphaeria* (68.5%). The bacterial genus *Pseudomonas* was found to be prevalent in 97.4% of the samples, followed by *Sphingomonas* (93.5%). *Fronfihabitans* and *Aeromicrobium* were present in 72.2% and 64.9% of all samples, respectively.

#### 2.3.4. Fungal and Bacterial Diversity and Composition in Organic and Conventional Apples at the Different Postharvest Steps

The disparities in the diversity and taxonomy of the fungal and bacterial microbiota altered by long-term storage and the different sampling steps were assessed for the two cultivation systems (organic and conventional). Regarding the fungal microbiota in the organic apple samples, the mean number of observed ASVs, based on the rarefied ASV table, was 23.83 on the surfaces of apples picked immediately after harvest. After 6 months of storage, this number almost doubled and reached 58.0. This increase was statistically significant (ANOVA, *p* < 0.001). The number of fungal ASVs reached 52.5 after the calibration line, 56.0 after 10 days of storage at 4 °C and finally 45.7 after a period of storage at 4 °C for 10 days followed by a period of deck storage of 4 h at ambient temperature. The results of the observed ASV numbers for conventionally grown apples had a similar trajectory as for the organically grown apples. Immediately before storage, the number of observed ASVs was about 34.6; it reached 56.8 after 6 months of cold storage under CA and 57.62 after passing through the calibration line. This increase was also statistically significant (ANOVA, *p* < 0.001). It then reached 48.7 at the last sampling step (Appendix A). Therefore, 6 months of cold storage under CA had a significant effect in terms of increasing the richness of the fungal communities (number of taxa) on the surfaces of the sampled apples. Deck storage seems to decrease the richness of the fungal communities. Moreover, the comparison of the Shannon diversity of organically and conventionally grown apples indicated that the fungal diversity was not statistically significant at the different sampling steps, while taking into consideration the method of cultivation (*p* > 0.05) (Figure 2A). However, samples from the last postharvest step (before transport) had the lowest fungal diversity, followed by the samples at the beginning of storage. Moreover, the postharvest stage had a significant effect on the community composition of the tested apples for conventional and organic apples. The PCoA analysis based on the Bray–Curtis dissimilarity test (Figure 3) showed clustering in the fungal communities, especially on the surfaces of organic apples picked directly after harvest, in comparison to the rest of the sampling steps. The PERMANOVA analysis indicated that the different postharvest sampling steps (R^2^ = 0.12, *p* < 0.001) and the cultivation techniques (R^2^ = 0.04, *p* < 0.001) had significant effects on the composition of the fungal epiphytic communities. The apples picked before a period of 6 months of cold storage appeared to harbor a distinct fungal community in comparison with the rest of the sampling steps. Moreover, the fungal community at the first sampling step was significantly different between organically and conventionally grown apples (Appendix A). Furthermore, the different sampling stages in the packinghouse after the period of cold storage appeared to produce similar fungal communities. These results were more evident for the organically grown apples (Figure 3B), confirming the impact of the storage conditions in the modification of the microbial ecosystems. 

Moreover, the results of the most abundant 12 species in the different samples (Figure 4), picked at the postharvest steps and for both cultivation techniques, support the observations cited above. For conventionally grown apples, at the beginning of storage, the most abundant species was *Aureobasidium pullulans* and species from the genera *Alternaria* sp., *Cladiosporium* sp., *Entyloma* sp. and *Sporobolomyces* sp. After 6 months of storage, all species except two were no longer present among the top 12 species, and new ones appeared, such as *Cystobasidium pinicola*, *Cadophora luteo-olivacea*, *Monocillium mucidim* and *Vishniacozyma victoriae*. In the post-calibration step, new species appeared, such as *Botryotinia pelargonii* and one species of the order Entylomatales, and *P. expansum* appeared in one sample out of eight. In the last two steps, the diversity seemed to be reduced. In the post-conditioning step, a species of the genus *Ramularia* sp. appeared, with the reemergence of *Cladosporium* sp. and the prevalence of *Diplodia scrobiculata* in two samples. *P. expansum* was also abundant in this step. In the last step, *P. expansum* and *Cladosporium* sp. were the most abundant. *Monocilium mucidum* was also abundant in one sample.

For the organically grown apples, important differences in the diversity of the most abundant taxa were observed compared to the conventionally grown apples. In the orchard, the most abundant taxa were similar to the case of conventional apples, with the prevalence of *Aureobasidium pullulans* and species from the genera *Alternaria* sp. and *Cladiosporium* sp. It is important to note, at this step, the high homogeneity between the different samples, which seemed to be less important in the following steps. *P. expansum* appeared among the 12 most abundant species after 6 months of storage on the surfaces of organic apples. Moreover, at this step, *Cyberlindnera misumaiens*, *Zygotorulaspora florentina* and species of the order Hypocreales were most abundant, but these two species were absent on the surfaces of organically grown apples. *Zygotorulaspora florentina* and *Cladiosporium* sp. persisted in the rest of the sampling steps. *P. expansum* reappeared in the step before transportation, when the apples underwent a period of deck storage. At this step, also, changes in fungal diversity were observed, with the appearance of a species of the family *Sclerotiniaceae.*

Regarding the bacterial diversity in organically and conventionally grown apples, the mean values of the observed ASVs were significantly different between the different sampling steps (ANOVA, *p* < 0.001) and cultivation systems (ANOVA, *p* < 0.05) (Appendix A). The richness of the bacterial communities on the surfaces of organic apples was statistically lower at the beginning of storage (mean value = 23.84); it reached a mean value of 64.84 after 6 months of storage. After passing through the calibration line, the mean value of the observed ASVs reached 92.11; this value was statistically significantly different from the one observed at the last step, after a period of deck storage (mean value = 34.67). Regarding the conventionally grown apples, the same trend was observed, with higher richness in the steps following long-term cold storage and a statistically significant decrease in richness at the steps preceding the transport of apples to the processing industry. The diversity of bacteria within the apple samples was also evaluated using the Shannon diversity index (Figure 2B). Organic and conventional apples from the category “before storage” showed the lowest bacterial diversity (Shannon index = 1.94 and 2.15, respectively). The fungal diversity was highly increased after 6 months of cold storage and continued to increase in the following postharvest steps. It was significantly reduced in the last two steps, i.e., after a period of storage for 10 days at 4 °C followed by deck storage at ambient temperature. The observed differences in diversity among sampling stages were due to changes in the presence/absence of taxa in organically and conventionally grown apples and/or significant changes in the relative abundance of specific taxa. Moreover, the beta diversity analysis based on the Bray–Curtis distance matrix indicated clear clustering between apples before and after long-term cold storage and the different postharvest stages (Figure 5). Both the sampling step (R^2^ = 0.25, *p* < 0.001) and the method of cultivation (R^2^ = 0.03, *p* < 0.001) had a significant effect on the composition of the epiphytic bacterial communities on the apples. In organically grown apples, clear clustering was observed between apples picked immediately after harvest and on the same apples after undergoing long-term cold storage (6 months). Furthermore, clustering was also highlighted in conventionally grown apples, where apples picked at the beginning of storage, after calibration and after conditioning had statistically significant differences in their epiphytic bacterial communities. The PCoA of the bacterial populations associated with the surfaces of conventionally and organically grown apples divided per sampling step are presented in Appendix A. The PERMANOVA test showed statistically significant differences in the compositions of the bacterial communities between the two cultivation systems (*p* < 0.001). Clear clustering between organic and conventional apples was observed at the step after long-term storage and post-conditioning, as well as overlapping after 4 h of deck storage.

Moreover, the relative abundance of the top 10 species in each step and for both organically and conventionally grown apples is presented in Figure 6. A species of the genus *Sphingomonas* was found in the top 10 most abundant species in all samples. For conventionally grown apples, at the beginning of storage, the following species were the most abundant: *Clavibacter* sp., *Erwina billingiae*, an unidentified species of the family *Enterobacteriaceae*, *Pantoea* sp., *Pantoea vagans*, *Gluconobacter* sp. and an unidentified species of the genus *Sphingomonas*. After six months of cold storage, new species emerged, such as *Luteibacter Rhizovicinus,* an unidentified species of the family *Microbacteriaceae*, *Methylobacterium* sp., *Rhizobium soli* and *Rhodococcus* sp., while others disappeared from the top 10 species, such as *Pantoea* sp., *Pantoea vagans* and an unidentified species of the family *Enterobacteriacies.* At the following steps, new species were also observed in the 10 most abundant species, such as *Pseudomonas* sp. after passing through the calibration line, *Frondihabitans* sp. after passing through the conditioning line and *Aeromicrobium ginsengisoli* and *Gluconobacter cerinus* at the last step. Regarding organic apples, similar results as for the conventional apples were observed; long-term storage modified the composition of the 10 most abundant species, with the absence of certain taxa after a period of storage, such as Pantoea sp., *Pantoea vagans* and an unidentified species of the family *Enterobacteriacies*, and the presence of new taxa, such as *Gluconobacter cerinus* and *Luteibacter Rhizovicinus.* Compared to the conventionally grown apples, the composition of the 10 most abundant species in the organically grown apples was different.

The differences between the most abundant bacterial and fungal species found in samples contaminated with patulin and in samples not exhibiting levels of patulin contamination were examined in conventional and organic apples. Figure 7 shows the relative abundance of the top 10 species in contaminated and non-contaminated samples. The composition of the 10 most abundant species was different between apples containing patulin and sound apples. Naturally, *P. expansum* was among the most abundant species in patulin-contaminated samples and was absent in the other group. For both conventional and organic apples, *Aureobasidium pullulans* and *Cyberlindnera misumaeiensis* were also present among the most abundant species in contaminated apples. In only conventional apples, contaminated apples also exhibited *Vishniacozyma victoriae* on their surfaces, *Monocilium mucidum* and a species of the genera *Cladosporium* sp. As for the organically grown apples, a species of the family *Sclerotiniaceae* was also abundant. In non-contaminated apples, both organically and conventionally grown apples exhibited on their surfaces species not present among the most abundant species of contaminated apples, such as a species of the genus *Alternaria* sp., *Cadophora luteo-olivacea*, *Cystobasidium pinicola* and a species of the genus *Sporobolomyces* sp. Other species were the most abundant in organically grown non-contaminated apples: *Gibberella baccata*, *Zygotorulaspora florentina* and a species of the order Hypocreales. 

#### 2.3.5. Association Networks among Fungal and Bacterial Species

To explore potential interactions among the various species present on the surfaces of the apples, we constructed fungal and bacterial co-occurrence networks for species appearing in at least five samples using the SPIEC-EASI pipeline [25]. These co-occurrence networks may indicate cooperative or competitive interactions. Figure 8 illustrates the bacterial and fungal species demonstrating positive interactions on the surfaces of the sampled apples, while Figure 9 depicts taxa displaying negative interactions on the apple surface. This analysis aids in identifying potentially beneficial relationships among different fungal and bacterial species within the microbial community, including interactions with *P. expansum*. The information on competitive interactions helps to identify specific species that can be tested in vitro for their potential biocontrol effects against pathogenic fungi. A total of 109 cooperative interaction pairs and 5 pairs of competitive relationships were identified.

Co-occurrence was observed between *P. expansum,* known as a psychrophilic fungus, and the psychrophilic yeast species *Mrakia frigida*. The latter was also positively correlated with *Tausonia pullulans*, another psychrotolerant yeast. *P. expansum* had also a strong positive interaction with a species belonging to the genus *Gluconobacter* sp., which had also a co-occurrence association with *Gluconobacter cerinus* and *Cyberlindnera misumaiensis*. These fermentative bacteria and yeasts could play a role in apple cell degradation, increasing the contamination of *P. expansum*. Moreover, a correlation matrix was constructed based on all observed ASVs from the microbiome composition dataset (multiple ASVs could be assigned to the same taxa), and it showed positive interactions between *P. expansum* and two other yeast species, *Mrakia blollopis* and *Cyberlindnera missouriensis.* The latter was also positively correlated with *Vishniacozyma tephrensis*. Moreover, *Mrakia frigida* was found to be positively correlated with *Tausonia pullulans*, which confirmed the positive interaction of *P. expansum* with psychrophilic yeasts (Appendix A). 

Furthermore, a negative correlation was found between *P. expansum* and *Cystobasidium pinicola* which was also negatively correlated with *Exobasidium* sp. (Appendix A). These negative interactions with *P. expansum* could be studied with the aim of biocontrol. 

Five pairs of fungal and bacterial competitive relationships were also obtained, including two with other apple pathogens (Figure 9). *Sporobolomyces* sp. was found to be negatively correlated with the pathogenic fungus *Nofabraea vagabunda*, responsible for the postharvest disease of apple bull’s eye rot. The fungal species *Epicoccum nigrum* exhibited negative interactions with the pathogenic species *Cadophora luteo-olivacea,* a postharvest pathogen responsible for the side rot of apple. Two bacterial species, *Pseudomonas frederiksbergensis* and *Pseudomonas libanesis*, were negatively correlated with the pink yeast *Sporobolomyces roseus*. 

## 3. Discussion

Domesticated apples (*Malus x domestica*) are regarded as an important crop worldwide as they can be stored for a long period of time, ensuring an appropriate supply of fresh fruit throughout the year [3,4,26]. However, apples are susceptible to infection by pathogens and therefore the development of decay during long-term postharvest storage [5]. Moreover, a period of cold storage followed by transfer to warmer temperatures has been proven to stimulate the development of decay on fruits and therefore mycotoxin production [27].

In this study, the work was focused on the investigation of the effect of multiple postharvest stages on the incidence of patulin contamination in apples of the variety “Golden Delicious”, destined for the production of compotes. This type of apple undergoes long-term cold storage under CA (up to 12 months in the most extreme cases). This study included conventionally and organically grown apples stored for 6 months at 1 °C with O_2_ and CO_2_ levels, respectively, at 0.9% and 1%.

### 3.1. Effect of Postharvest Steps on P. expansum and Patulin Occurrence

After harvest, patulin was detected in two out of nine conventional apple samples and *P. expansum* was also quantified on five out of nine samples. For organic apples, immediately after harvest, no patulin was detected and *P. expansum* was only quantified in one sample. After 6 months of cold storage at 1 °C under CA, the incidence of patulin and fungus contamination in conventional apples was the same; however, the organic apple samples showed important levels of contamination by the mycotoxin, with a significant increase in the number of contaminated samples by *P. expansum*. A study carried out by Dos Santos et al. showed that the use of CA conditions in storage was not effective in preventing patulin accumulation in apples [28]. Moreover, *P. expansum* is a psychrophilic fungus, meaning that it can grow well at 0 °C or less (−2/−3 °C); thus, the infection of apples by this fungus can still take place during cold storage [29]. The efficiency of cold storage in controlling the accumulation of the mycotoxin has also been examined and is known to be controversial. Jackson et al. indicated that patulin production occurred during cold storage [30], and Morales et al. proposed that its production was only possible when *P. expansum* reached a critical mycelial mass during cold storage [31]. Moreover, the firmness of the apple flesh is associated with resistance against blue mold disease and patulin production. In fact, the risk of pathogen infection becomes higher as the fruit ripens and the flesh softens [32]. A study by Vilanova et al. in 2014 on “Golden Smoothee” apple samples demonstrated increased susceptibility to decay as the fruit ripened [33]. Moreover, the patulin content was positively correlated with the loss of the firmness of the fruit [34]. This correlation may elucidate the observed high levels of patulin in organic apples post-storage, attributed to the loss of fruit firmness. Following this, in both organic and conventional apple samples, the different procedures in the packinghouse (calibration and conditioning) had no effect on the incidence of *P. expansum* contamination, with high levels detected on the surfaces of the analyzed apples. After this, the apples underwent a period of deck storage at ambient temperature, before the loading of the transportation bin. This step is critical for patulin contamination and *P. expansum* infection. An increase in the number of contaminated apples by the mycotoxin was observed for both conventional and organic apples. In a study carried out in 2018 by Gougouli et al., it was demonstrated that, at high storage temperatures ranging between 20 and 25 °C, *P. expansum* experienced faster sporulation and higher mycelia growth [35], explaining the high levels of patulin detected. In addition, the period of deck storage following cold storage had a significant effect on the infection of organic apples by *P. expansum* (100% of the tested samples were positive for the fungus). Morales et al. (2010) concluded that when cold storage (even if treatments were applied at this step to prevent fungal contamination) is followed by ambient storage, the growth of the fungus is reactivated, and the percentage of rotten apples can reach similar rates to the case of untreated apples [36]. Moreover, these results imply a potential risk of patulin contamination in apple-based products when the apples used undergo extended storage periods followed by a brief period of deck storage. This is concerning because once patulin is produced in rotting apples, there is a high likelihood of it being present in the final product, as the processing steps may not be sufficient to eliminate this mycotoxin. In a recent study carried out by Tangni et al. in 2023 regarding patulin levels in apple-based products purchased on the Belgian market, patulin was detected in 54.4% of the examined apple juices, reaching concentrations of up to 191.1 μg/L, and in 7.1% of the puree samples, reaching 35.9 μg/kg [37]. 

Moreover, no correlation between the fungus concentration and patulin production was identified in both organic and conventional samples. The presence of the fungus was not an indicator of the level of contamination of apples by patulin. Dos Santos et al. (2022) showed no correlation between the patulin and fungal concentrations, indicating that high concentrations of this mycotoxin can be detected even in samples with a low fungal presence and vice versa [38]. Furthermore, the variability among strains could be a significant factor affecting the accumulation of patulin during the extended storage of apples [39]. Moreover, the incidence of patulin in organically grown apples was not significantly higher than for conventionally grown apples. These results coincide with other studies conducted on conventionally and organically grown apples, where the cultivation system had no effect on the increase in patulin levels in the fruit [40,41].

The epiphytic microbiomes of fruits encompass a diverse array of microorganisms that influence the health, quality and disease resistance of the fruit throughout both the pre-harvest and postharvest stages. It is also involved in many of its host functions, affecting not only the physiology and biochemistry of the host but also its growth and stress tolerance [22]. Therefore, the study of the fruit microbiome plays an important role in the management and control of pre- and postharvest pathogens [20,21,22]. In a recent study carried out by Droby et al. (2022), postharvest disease development in apples was explained through a complex interaction between the pathogen and specific microorganisms located at the level of the wound, rather than simply the action of a single pathogen [24]. In this study, the effect of long-term cold storage was evaluated, as well as multiple postharvest steps, on the diversity and composition of the epiphytic fungal and bacterial communities of organically and conventionally grown apples. The changes in the structure of the microbial communities and their relationship with the pathogenicity of *P. expansum* were also examined. In addition, specific taxa were identified that co-occurred on the surfaces of apples with *P. expansum*, as well as others that showed competitive effects against this fungus.

### 3.2. Apple Core Microbiome

The global core microbiome of “Golden Delicious” apples (organic and conventional) was examined. The global core microbiome was determined by the taxa present in at least 75% of all samples [19]. The seven fungal genera that constituted the core microbiome of all studied apples were *Aureobasidium*, *Vishniacozyma*, *Cladosporium*, *Leptosphaeria*, *Alternaria*, *Cadophora* and a non-identified genus, and the seven bacterial genera were *Pseudomonas*, *Sphingomonas*, *Clavibacter*, *Methylobacterium*, *Pantoea*, *Rhizobium* and a non-identified genus. *Sphingomonas* is a Gram-negative, aerobic bacterial genus, linked to the promotion of plant growth due to its capacity to produce gibberellins and indole acetic acid in response to different abiotic stress conditions [42]. These phytohormones are also involved in fruit maturation, development and quality. Moreover, *Methylobacterium* has plant growth-promoting properties [43]. Similar results were observed in a study carried out on “Royal Gala” apples, exhibiting high similarity regarding the composition of their core microbiome [19]. A study on the apple endophytic microbiota of different scions found that the compositions of the fungal taxa of “Golden Delicious” and “Royal Gala” were very similar to each other due to the fact that these two scions were closely related by pedigree [44]. In addition, *Aureobasidium* and *Cladosporium* have been reported to be involved in the microbiome of apple [19,45,46]. The core microbiome can potentially play an important role in defining the apple’s traits regarding its quality and resistance to disease.

### 3.3. Effect of Postharvest Steps on Fungal and Bacterial Communities

The alterations in the fungal and bacterial communities in conventional and organic apples were assessed at the different postharvest stages. After 6 months of cold storage, the richness of the fungal communities was significantly higher, and it decreased after deck storage at ambient temperature. No differences were observed between conventional and organic apples regarding the richness of the fungal taxa. The Shannon diversity index showed no differences in the fungal diversity of conventional and organic apples in all postharvest stages. However, the beta diversity analysis showed clear clustering between the microbial composition of apples immediately after harvest and that after 6 months of storage. Moreover, the fungal community at the first sampling step (immediately after harvesting) was significantly different between organically and conventionally grown apples. Furthermore, the different sampling stages in the packinghouse after the period of cold storage appeared to produce similar fungal communities. These results were more evident for the organically grown apples. The organic apples picked before a period of 6 months of cold storage appeared to harbor a distinct fungal community in comparison with the rest of the sampling steps. The most dominant taxa (in both conventional and organic apples) were *Aureobasidium pullulans*, a species belonging to the genus *Cladosporium* sp. and a species belonging to the genus *Alternaria* sp. *A. pullulans* is a yeast-like fungi that was isolated from the surfaces of apples and exhibits antagonistic activity against the growth of *P. expansum* [47]; it has even been used to develop a bio-fungicide used against blue mold disease at pre-harvest stages [48]. Both *Aureobasidium* and *Alternaria* were abundant on apple samples picked before a period of long storage [45]. The variability of the fungal composition among the different apple samples picked at the same step was apparent after long-term storage under CA, whereas all samples picked directly after harvest were highly homogeneous. A clear shift in the fungal communities was observed after the storage period, featuring new taxa and a decrease in the abundance of other taxa, such as *Alternaria* sp.; these results coincide with those obtained by Wassermann et al., suggesting that the reduction was due to the cold sensitivity of these taxa [45]. The new most abundant taxa included *Cadophora luteo-olivaceae*, which was consistent across all postharvest stages, in both organic and conventional apples; it is a pathogenic fungi known to cause postharvest disease in grapes and kiwifruit [49,50]. A recent study carried out by Carneiro et al. confirmed the virulence of this fungus on apples of the variety “Golden Delicious” and highlighted the importance of raising awareness about this potential apple disease, which causes considerable losses to other crops [51]. In regard to *P. expansum*, it appeared in the most abundant taxa after long-term storage and exhibited an increase in relative abundance over the storage period in both organic and conventional apples. This could have been due to the aging of apples and the release of exudates on the surfaces of apples, which support the proliferation of this pathogen. These results were also demonstrated by Abdelfattah et al. [52]. In conventional apples, the fungus *Cystobasidium pinicola* appeared in the most abundant species after 6 months of storage and was consistent throughout the rest of the sampling steps in conventionally grown apples. This yeast species was tested against *Hymenoscyphus fraxineus*, a pathogenic fungus, and exhibited antifungal actions in vitro by the formation of an inhibition zone [53]. The composition of the most abundant fungal taxa was very similar between organic and conventional apples, with two species only present on organically grown apples: *Botryotinia pelargonii* and *Diplodia scrobiculata*. 

The richness of the bacterial diversity was affected significantly by the sampling steps and the cultivation system. Long-term cold storage increased the richness of the bacterial community, and deck storage following the storage of apples at 4 °C significantly decreased the richness. The lower diversity at this stage compared to that in the rest of the sampling steps might indicate lower stability and a decrease in the health of the ecosystem [54]. Long term-cold storage followed by a calibration step in the packinghouse had a significant effect on the bacterial diversity of both conventional and organic apples, which was greater than the one observed immediately after harvest, and it significantly decreased after the step of deck storage. Wassermann et al. also observed that the bacterial diversity was lower before storage in apples [45]. Moreover, both the sampling steps and the method of cultivation had a significant effect on the composition of the epiphytic bacterial communities on apples. The observed differences in diversity among the sampling stages were due to changes in the presence/absence of taxa in organically and conventionally grown apples and/or significant changes in the relative abundance of specific taxa. In fact, the composition of the most abundant species on the surfaces of apples was affected by the different sampling steps. Immediately after harvest, in both organic and conventional apples, the most abundant species were similar. 

It was also observed that the majority of taxonomic representation in organic and conventional treatments was shared for both bacterial and fungal taxa, especially in the step before cold storage. This was also demonstrated by Ottesen et al., suggesting that the environmental parameters may have a strong influence on the epiphytic microbiota despite the different management protocols [55].

The most abundant taxa in conventional and organic apples were also observed in samples contaminated with patulin and non-contaminated samples. Species belonging to the genus *Alternaria* were abundant in both contaminated and non-contaminated apples. Therefore, there is a high risk of contamination of these apples by mycotoxins produced by *Alternaria* sp., even in samples that are not contaminated with patulin. The fungal species *Cyberlindnera misumaiensis* was present alongside *P. expansum* exclusively in the samples contaminated with patulin.

Moreover, inter-species interactions are very important in shaping fungal and bacterial dynamics. The different network analyses presented in this work demonstrated intra-species relationships. Multiple pairs of fungal and bacterial competitive relationships were observed. A species belonging to the genus *Sporobolomyces* was found to be negatively correlated with the pathogenic fungus *Nofabraea vagabunda*. The latter causes bull’s eye rot, one of the main postharvest diseases of apples [56]. The fungal species *Epicoccum nigrum* exhibited negative interactions with the pathogenic species *Cadophora luteo-olivacea*. In fact, *E. nigrum* is known for its biological control use against the postharvest brown rot caused by *Monilinia* spp. [57]. As already mentioned above, *C. luteo-olivacea* is a postharvest pathogen causing side rot in apples in packinghouses [51]. Moreover, a negative correlation was found between *P. expansum* and *Cystobasidium pinicola*. As already mentioned above, this yeast species exhibited antifungal actions in vitro against *H. fraxineus*, a pathogenic fungus [53]. *Cystobasidium pinicola* was also found in the most abundant taxa of conventional apples not contaminated with patulin and was absent in contaminated apples. These results highlight the potential action of protective microbiota against pathogenic fungi on the surfaces of apples in the postharvest stages. In fact, the inter-species interactions obtained in our study require further investigation and verification. The network analysis performed in this study provides a basis for the further analysis of fungal and bacterial interactions for fruit disease biocontrol. 

Furthermore, positive interactions were observed between *P. expansum* and multiple fungal and bacterial species. These co-occurrence associations could be indicative of cooperative interactions with the pathogenic fungus. The network analysis showed a positive interaction between *P. expansum* and the psychrophilic yeast species *Mrakia frigida*. The latter was also positively correlated with *Tausonia pullulans*. *P. expansum* had a strong positive interaction with a species belonging to the genus *Gluconobacter* sp., which had also a co-occurrence association with *Gluconobacter cerinus* and *Cyberlindnera misumaiensis*. *G. cerinus* exhibits antifungal activity against table grape rot pathogens [44]. *P. expansum* exhibited also positive interactions with two other fungal species, *Mrakia blollopis* and *Cyberlindnera missouriensis*. The latter was absent in the most abundant species directly after harvest and was abundant in the stages following long-term cold storage, in both organically and conventionally grown apples. It was also present in abundance on the surfaces of apples contaminated with patulin and absent in non-contaminated apples. 

## 4. Conclusions

This study was focused on the investigation of the effect of multiple postharvest stages on the incidence of *P. expansum* and patulin contamination in apples of the variety “Golden Delicious”, organically or conventionally cultivated and destined for the production of compotes. 

In general, the incidence of patulin and the occurrence of *P. expansum* on the surfaces of apples was not significantly different between the two cultivation techniques. Patulin production occurred during cold storage and the CA conditions were not effective in preventing *P. expansum* growth and patulin accumulation in apples. The increase was less significant for conventional apples as they showed important levels of contamination prior to the storage period. The period of deck storage was critical for both *P. expansum* and patulin contamination. An increase in the number of contaminated apples by the mycotoxin was observed for both conventional and organic apples.

The effect of the postharvest stage on the diversity and composition of the epiphytic fungal and bacterial communities of organically and conventionally grown apples was examined. It was observed that most of the taxonomic representation in the organic and conventional treatments was shared for both bacterial and fungal taxa in the step before cold storage. A clear shift in the fungal communities was observed after the storage period (especially for organic apples), featuring new pathogenic taxa alongside *P. expansum*. *Cyberlindnera misumaiensis* was also positively correlated with *P. expansum* in the network analysis, which could indicate a cooperative interaction with *P. expansum.*

Many significant positive interactions between different species were detected in the network analysis. Some negative associations have also been addressed. These associations can be used to develop predictions of potential future biocontrol agents against pathogens. The findings of this investigation provide a comprehensive view of the postharvest apple microbiome, laying the groundwork for an enhanced understanding of the potential interactions within the microbial community of the fruit.

## 5. Materials and Methods

### 5.1. Apple Sampling and Handling

Apples of the variety “Golden Delicious”, destined to produce apple compotes, grown in Moissac, France, were selected for this study. Two parcels were chosen for the follow-up of both organically and conventionally grown apples. All apples were harvested manually in September of 2018 and kept in plastic storage bins (dimension: 1 m^3^). Three storage bins per cultivation technique were marked and followed up from the orchard until the transportation of the apples to the processing industry.

After harvest, different storage systems can be used for the conservation of apples [45]. In this study, the apples underwent long-term storage; they were stored at 1 °C for 6 months under a controlled atmosphere (CA) (O_2_ = 0.9% and CO_2_ = 1%). After 6 months of storage, the apples were removed from the CA storage rooms and were kept at 4 °C in the station. The apples were passed through the calibration line by dipping the storage bins into a water bath and sorting the apples by caliber; they were graded using an automated apple sorting system. This step lasted approximately 2 h at 4 °C.

In the context of this study, the apples were collected in their respective storage bins without taking into consideration their different calibers, to allow the follow-up of the same batch in the remaining steps. Apples were sampled after CA storage, immediately before calibration, and at the end of the calibration line. Then, the apples were stored at 4 °C for 10 days before packaging. In the context of our study, the apples were collected in their respective storage bins and sampled at the end of the packaging line (apples were kept at 4 °C for 10–15 days before transport). Then, the apples were stored at ambient temperature (approximately 25 °C) for 6 h and loaded into the transporting bin. This storage period is referred to as “deck storage”.

At each step, the apples were sampled from their storage bins as follows: in order to cover all bins, three sampling levels were assigned by bin and 5 apples were randomly picked at each level (from 3 storage bins) and kept in groups of five in sterile polyethylene bags. An overview of the sampling steps is illustrated in Figure 10, with a visual representation of the storage bins and the sampling levels per bin. Therefore, for each sampling step, 45 conventionally and 45 organically grown apples were analyzed. Moreover, 100 mL of water from each water bath and flume water (calibration line and packaging line) was sampled in triplicate for analysis. 

In all subsequent analyses, the statistical significance of the variations among samples selected at three bin levels was assessed through one-way ANOVA, followed by Tukey’s test (Tukey’s HSD, *p* < 0.05), using the Statistica V.13 software (Statsoft Inc., Tulsa, OK, USA). No significant differences were identified among the distinct levels. Therefore, the results are presented without accounting for these variations.

### 5.2. Sample Preparation and Patulin Extraction from Apples

At reception, the apples were washed in groups of 5 by adding 250 mL of a washing solution to the bag and manually scrubbing the apples against the bag for approximately 5 min. The washing solution comprised 0.15 M NaCl supplemented with 0.1% Tween 20 (*v*/*v*) in distilled water, filtered under sterile conditions through a PTFE filter with a diameter of 0.2 µM. Then, 50 mL of the resulting washing solution underwent centrifugation at 4500× *g* for 10 min at room temperature and the pellet was stored at −20 °C until further analysis. The apples were transferred into clean bags and stored at −20 °C before analysis. An overview of the analysis carried out on one sample (composed of 5 apples) is illustrated in Figure 11.

Patulin extraction from the apples followed the method outlined by Al Riachy et al., with some adjustments [58]. Apples were squeezed using a laboratory paddle blender (Stomacher^®^ 400, Seward, Worthing, UK). Approximately 5 g of apple puree from each sample was measured into a 50 mL Falcon flask, followed by the addition of 750 µL of pectinase from Aspergillus aculeatus enzyme solution (SIGMA, Darmstadt, Germany) and 2.5 mL of H_2_O. The mixture was homogenized and left overnight at room temperature. Subsequently, the samples were centrifuged at 4500× *g* for 10 min, and the supernatant was transferred to a 50 mL Falcon flask, to which 10 mL of a solvent mixture (ethyl acetate/hexane, 60/40, *v*/*v*) was added. The samples were agitated for 10 min on a rotary agitator at 300 rpm. After this, 2 g of sodium bicarbonate was introduced, and the samples were agitated for an additional 10 min. To this mixture, 15 g of anhydrous sodium sulfate was added, and the samples experienced an additional round of agitation for 10 min. The resulting mixture was then centrifuged at room temperature at 3000 rpm for 10 min, and the entire supernatant (with volume noted) was collected in a 15 mL Falcon flask. This collected supernatant was evaporated in a SpeedVac Concentrator Plus (Eppendorf, Hamburg, Germany) for 60 min at 40 °C. The resultant dry extract was suspended in 1 mL of acidulated water (H_2_O/acetic acid, 99.5/0.5, *v*/*v*), and the samples were sonicated for 20 min to enhance the dissolution. The filtered samples were then passed through a 0.45 µm cellulose acetate (CA) syringe filter (Phenomenex, Torrance, CA, USA) into a clean 2 mL vial.

### 5.3. Patulin Quantification in Apples

The analysis of the patulin concentration in various samples was performed using the SHIMADZU HPLC system (Kyoto, Japan). This system is equipped with a UV detector and a LiChro-spher^®^ 5 µ ODS 250 × 4.6 mm C18 HPLC Column (Merck, Darmstadt, Germany), running at a flow rate of 1 mL/min and a temperature of 35 °C. The HPLC system received an injection of 100 µL of the extracted mycotoxin. The mobile phase consisted of ultra-pure water (Phase A) and acetonitrile (Phase B), following a gradient program starting at 95% A (0.01 min), reaching 98% A at 16 min, decreasing to 40% A at 20 min, and returning to 95% A at 26 min until 30 min. The detection of patulin occurred at a wavelength of 277 nm. For quantification, a calibration curve was constructed using a patulin standard (LIBIOS, Vindry-sur-Turdine, France) with concentrations ranging from 1.5 ng/mL to 250 ng/mL. Finally, the method’s limits of detection (LOD) and quantification (LOQ) were determined, resulting in LOQ 1 µg kg^−1^ and LOD 0.3 µg kg^−1^.

### 5.4. DNA Isolation from the Surfaces of Apples

The DNA was extracted from the pellet obtained from 50 mL of the apple washing solution (see Section 3.2) and the extraction was performed using the FastDNA™ SPIN Kit (MP Bio-medicals, IIIkirch, France). The DNA extraction was conducted following the method illustrated by Al Riachy et al. [58]. The first step consisted of the resuspension of the pellet in 1 mL of the lysing solution CLS-TC (MP Biomedicals, Germany), included in the kit, and then its transfer to the lysing matrix A tube. The Garnet matrix and a 1/4″ ceramic sphere were incorporated into the latter to aid in cell lysis. Subsequently, the samples underwent grinding using the FastPrep^®^ Instrument for a duration of 50 s at a speed of 6 m/s. The remaining steps of the extraction followed the manufacturer’s specifications (MP Bio-medicals, IIIkirch, France). Finally, the DNA was resuspended in 50 µL of DNase-free water (DES), and its purity ratio and concentration were assessed using a Nanodrop ND 8000 spectrophotometer (Thermo Fisher Scientific, Waltham, MA, USA).

### 5.5. Quantification of P. expansum on the Surfaces of Apples Using q-PCR

The quantification of *P. expansum* DNA was conducted through q-PCR, utilizing the primer set Pexp_patF_F/Pexp_patF_R (0.2 µM, Sigma-Aldrich, Saint-Quentin Fallavier, France), specifically designed from the patF gene associated with patulin biosynthesis [59]. To prepare the DNA samples, all samples were diluted at a ratio of 1:5 (*v*/*v*) in DNase-free distilled water. The reaction mixtures consisted of 10.5 µL SensiFAST SYBR^®^ No-ROX (Bioline, Paris, France), 0.42 µL of each primer (10 µM), 10.5 µL template DNA and PCR-grade water, adjusted to a final volume of 14.5 µL. The q-PCR reactions were executed in the LightCycler^®^ 480 Real-Time PCR System (Roche Applied Science, Mannheim, Germany) with the following cycling conditions: initial denaturation at 95 °C for 20 s, followed by 45 cycles of denaturation at 95 °C for 30 s, annealing at 63 °C for 30 s, and extension at 72 °C for 15 s. Following the last amplification cycle, a melting curve was created by initially heating at 95 °C for 5 s, followed by a temperature decrease to 65 °C for 1 min, and concluding with a cooling step at 40 °C for 30 s.

For each prepared 384-well plate, ten-fold dilutions of *P. expansum* pure genomic DNA (NRRL 35695, Northern Regional Research Laboratory, Peoria, IL, USA), with a known concentration, were incorporated to establish a standard curve, enabling the quantification of *P. expansum* DNA on the apple surfaces. The LightCycler^®^ 480 System software (Roche Applied Science, Mannheim, Germany) automatically determined the quantification values of the DNA and the threshold cycle (Ct) values. To identify or exclude any potential DNA contamination, a negative control containing DNase-free distilled water instead of the DNA sample was included in every plate. Statistical differences in *P. expansum* DNA abundance between sampling stages and cultivation systems were determined using the one-way ANOVA test. 

### 5.6. DNA Amplification and Sequencing

#### 5.6.1. Library Construction and Sequencing

The library preparation and sequencing procedures were carried out following the methodology detailed by Al Riachy et al. [58]. For bacterial amplification, 16S V3/V4 amplicons were generated using the primer set 341 F and 785 R. A pair of peptide–nucleic acid PCR blockers (PNA Bio Inc., Newbury Park, CA, USA) was introduced to minimize the amplification of plant chloroplast and mitochondrial sequences. The fungal internal transcribed spacer 2 (ITS2) region underwent amplification using the primer pair ITS86 and ITS4. Illumina adapters (www.illumina.com (accessed on 12 April 2022) were included in both primer sets for subsequent multiplexing. In the case of ITS amplicon generation, PCR reactions were performed in a total volume of 25 µL, comprising 12.5 µL of AmpliTaq Gold™ 360 Master Mix (Thermo Fisher Scientific), 0.625 µL of each primer (0.25 µM), PCR-grade water and 5 µL of template DNA (without replicates). For 16S, the PCR reactions were performed in 25 µL, including 12.5 µL of AmpliTaq Gold™ 360 Master Mix (Thermo Fisher Scientific), 0.625 µL of each primer (final concentration 0.25 µM), 1 µL of each PNA (0.5 µM), PCR-grade water and 5 µL of template DNA.

The reactions were incubated in a Mastercycler X50 (Eppendorf France SAS, Montesson, France) for 94 °C for 10 min, followed by 30 cycles of 95 °C for 30 s, 55 °C for 45 s, 72 °C for 30 s and a final elongation at 72 °C for 10 min for ITS amplicon generation. For prokaryote 16S amplicon generation, the following cycle conditions were used: 95 °C for 5 min, 30 cycles of 96 °C for 1 min, 78 °C for 5 s for PNA clamping, 54 °C for 1 min, 74 °C for 1 min and a final elongation at 72 °C for 10 min.

Following magnetic bead purification (Clean PCR, Proteigene, Saint-Marcel, France), the indexing PCR was conducted in a total volume of 18 µL. This included 5 µL of first-round PCR products, 9 µL of Phusion^®^ High-Fidelity PCR Master Mix (NEB, Evry, France), 2 µL of I5 index adapter and 2 µL of I7 index adapter. The cycling conditions were as follows: initial denaturation at 95 °C for 3 min, followed by 10 cycles of denaturation at 95 °C for 30 s, annealing at 55 °C for 30 s, extension at 72 °C for 30 s, and a final elongation step of 5 min at 72 °C. To enable the multiplexing of all samples in a single MiSeq run, a set of 384 in-house index pairs was employed. The final PCR products were combined and subjected to paired-end sequencing on a MiSeq Illumina sequencer using the MiSeq Reagent Kit v3 (600-cycle; Illumina, San Diego, CA, USA).

#### 5.6.2. Data Analysis and Statistics

The sequencing data underwent demultiplexing and trimming through cutadapt. Subsequently, the forward and reverse paired-end reads were merged and subjected to cleaning of chimeras and filtering; then, the taxonomy was assigned using a dada2-based workflow. For taxonomy assignment, the UNITE 8.2 [60] and RefSeq_RDP 16S databases were utilized for ITS and ADNr 16S, respectively. Alpha and beta diversity analyses of the population structure and composition were conducted using the phyloseq package (version 1.32.0) in R (version 4.0.0), after rarefaction to an even number of reads, specifically 5516 per sample for 16S and 10,032 per sample for ITS2. The alpha diversity between the different sampling steps was analyzed with Tukey’s HSD pairwise comparison (α ≤ 0.001). Beta diversity differences were assessed using a permutational ANOVA (PERMANOVA, adonis function) with 9999 permutations.

The statistical approach employed to infer microbial ecological networks from the amplicon sequencing datasets was SPIEC-EASI (SParse InversE Covariance Estimation for Ecological Association Inference). SPIEC-EASI utilizes algorithms for sparse neighborhood and inverse covariance selection to construct the networks [25].

## Figures and Tables

**Figure 1 toxins-16-00102-f001:**
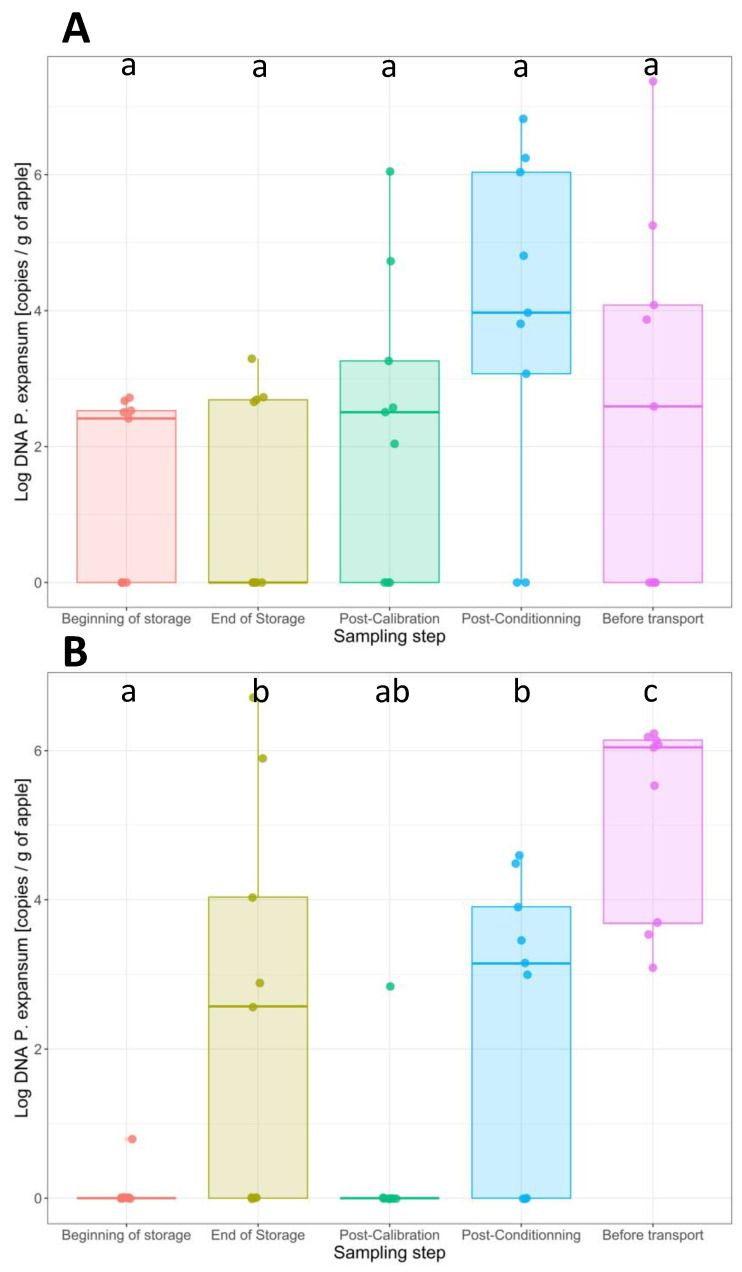
Boxplot representing the *P. expansum* DNA quantities (copies/g of apple) on the surfaces of 450 apples collected at various postharvest stages. The abundance of *P. expansum* within each stage was assessed through qPCR in three replicates. (**A**) DNA amounts of *P. expansum* on the surfaces of conventional apples. The levels were consistent between the three first steps and were more important in the last two steps. (**B**) DNA amounts of *P. expansum* in organic apples. Two steps significantly increased the quantity of *P. expansum* DNA on the surfaces of organically grown apples: the period of cold storage under CA for 6 months and the transfer of apples from 4 °C to an ambient temperature of 25 ± 5 °C (Tukey HSD test, *p* < 0.001).

**Figure 2 toxins-16-00102-f002:**
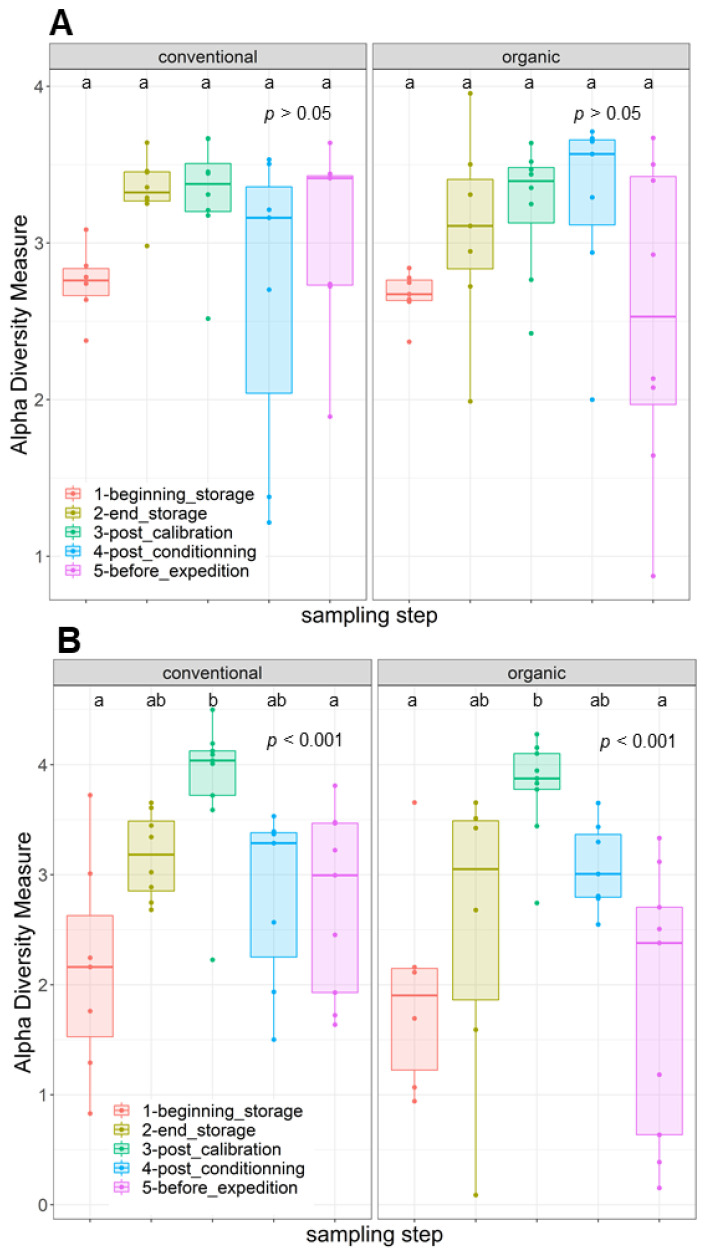
Boxplots depicting variance in Shannon diversity measures of (**A**) fungal and (**B**) bacterial communities on the surfaces of organically and conventionally cultivated apples across various postharvest stages. The distinct letters indicate significant differences among the groups in the analysis after conducting a Tukey’s HSD test.

**Figure 3 toxins-16-00102-f003:**
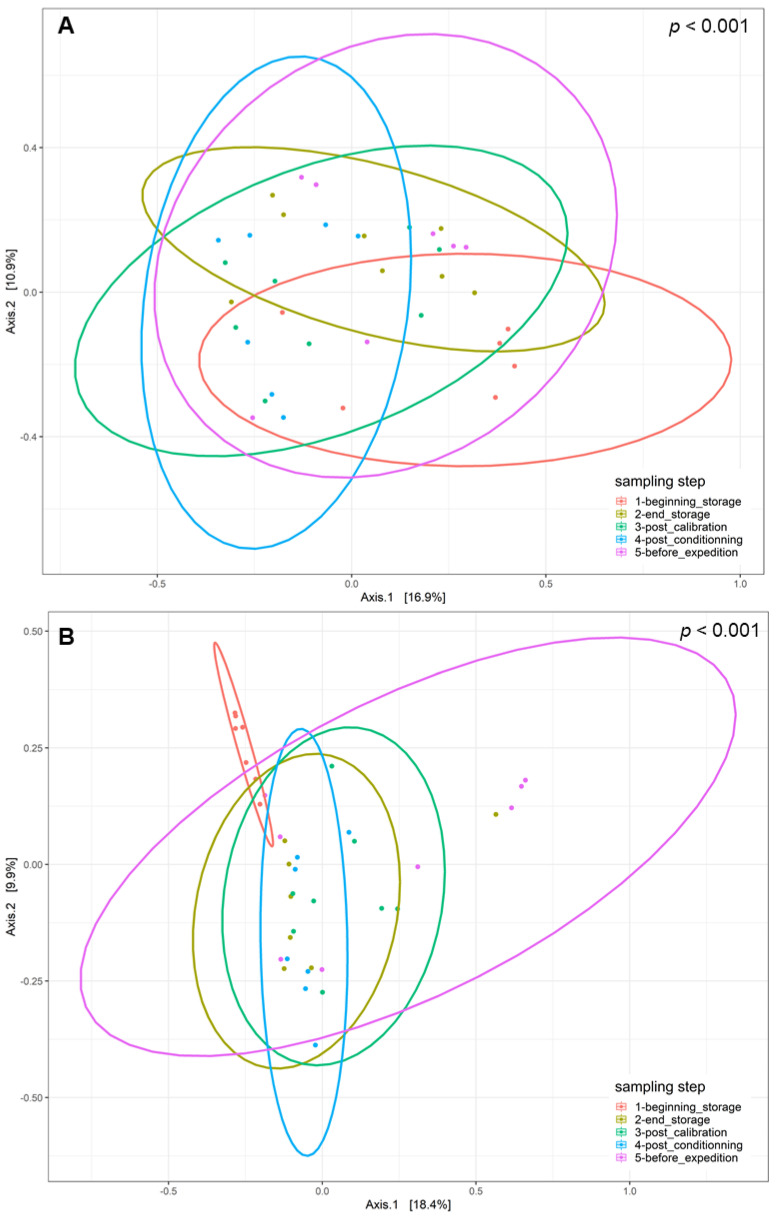
Principal coordinate analysis (PCoA) of fungal populations linked to the surfaces of (**A**) conventionally cultivated apples and (**B**) organically cultivated apples gathered at the five postharvest stages using the Bray–Curtis beta diversity metric.

**Figure 4 toxins-16-00102-f004:**
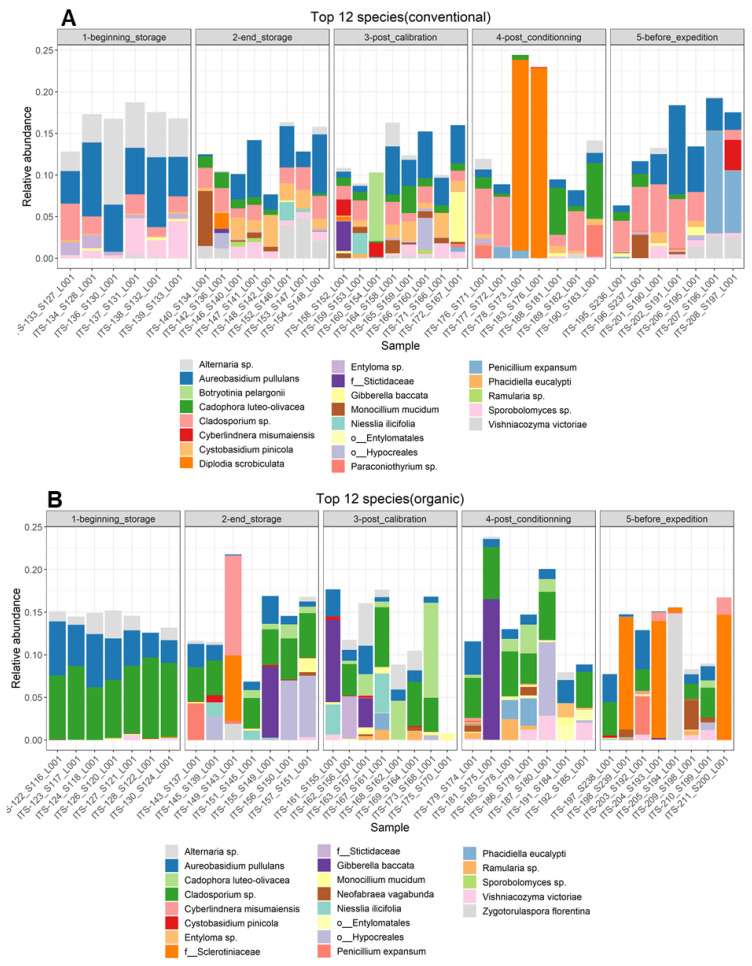
Relative abundance of the top 12 fungal species identified per sample on the surfaces of (**A**) conventionally harvested apples and (**B**) organically harvested apples at various postharvest sampling stages. In instances where taxonomic identification could not be performed to the species level, the ASV was identified at the lowest possible level on the phylogenetic tree.

**Figure 5 toxins-16-00102-f005:**
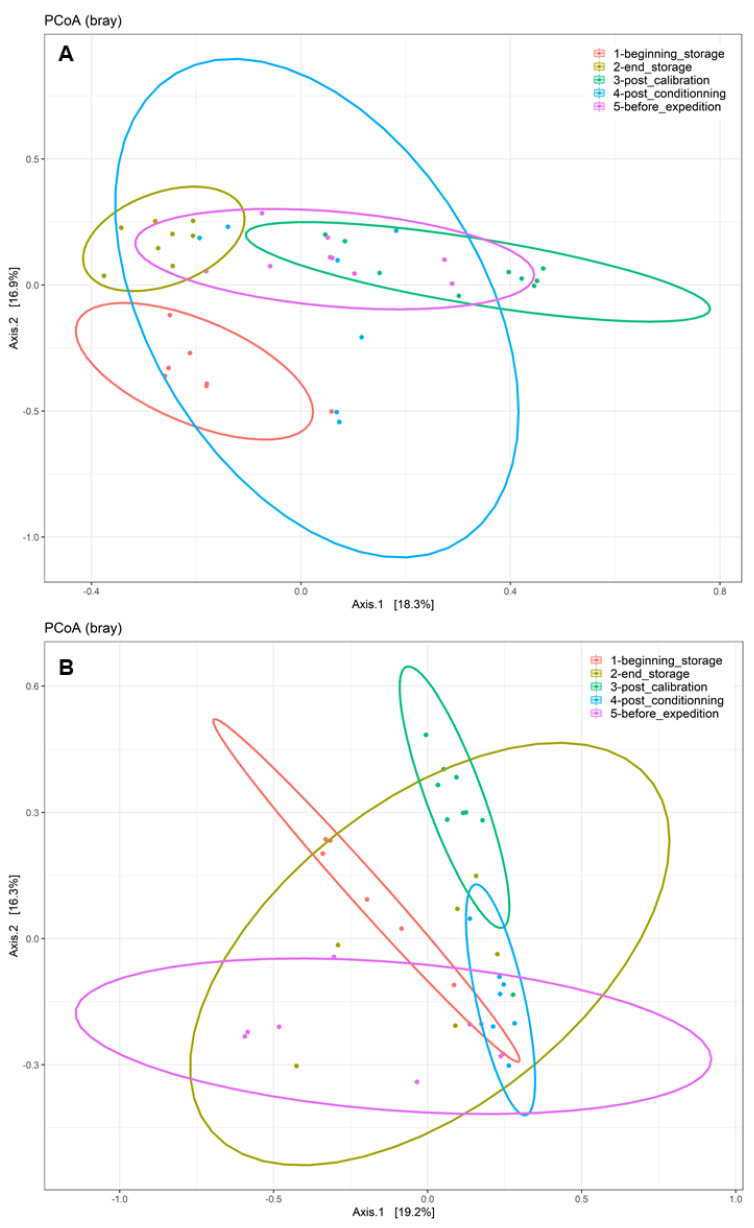
Principal coordinate analysis (PCoA) of bacterial populations linked to the surfaces of (**A**) conventionally cultivated apples and (**B**) organically cultivated apples collected at the five postharvest stages using the Bray–Curtis beta diversity metric.

**Figure 6 toxins-16-00102-f006:**
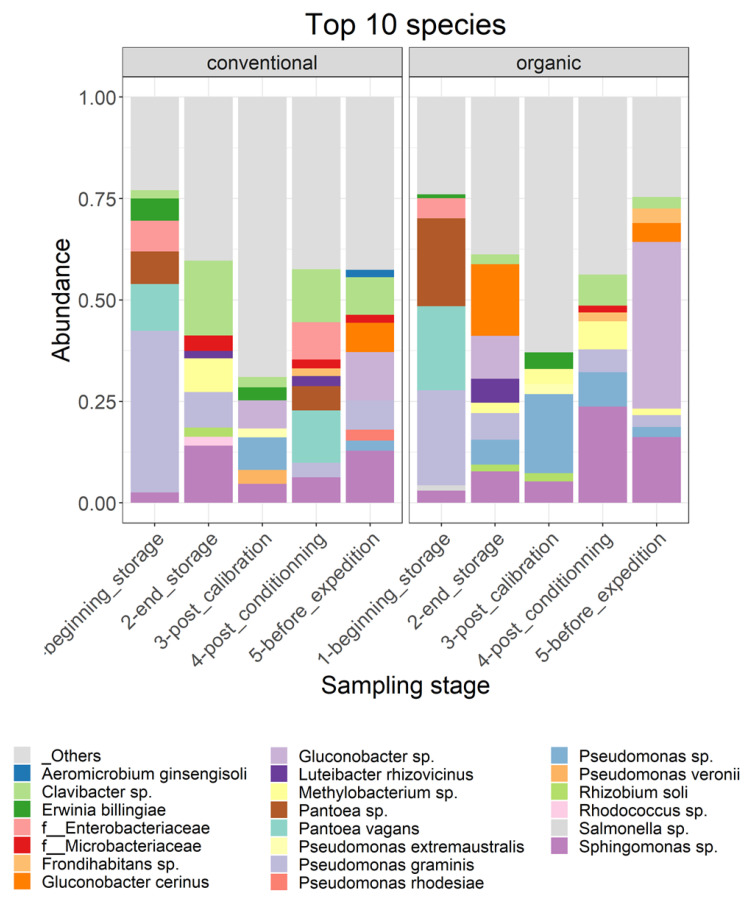
Relative abundance of the top 10 bacterial species identified per sampling step on the surfaces of conventionally harvested apples and organically harvested apples at various postharvest sampling stages. Bacterial species constituting less than 0.5% are categorized as “others”. In cases where taxonomic identification could not be established at the species level, the ASV was identified at the lowest possible level on the phylogenetic tree.

**Figure 7 toxins-16-00102-f007:**
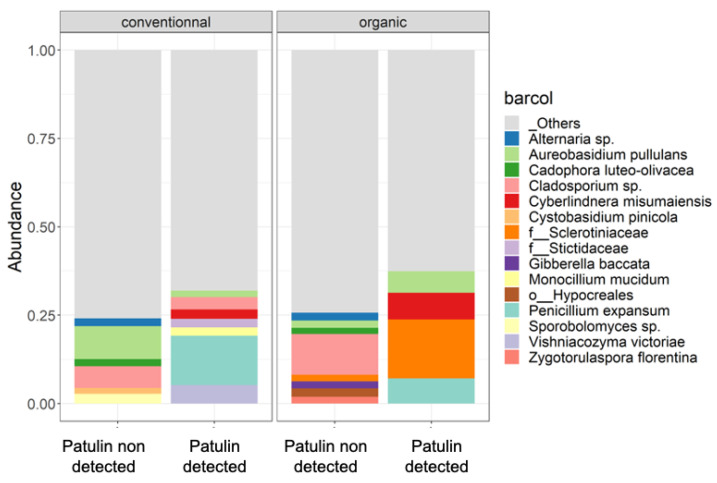
Relative abundance of the top 10 fungal and bacterial species found on the exteriors of conventional and organic apples when patulin was identified and when it was not detected in the samples by HPLC. Bacterial species constituting less than 0.5% are categorized as “others”. In instances where taxonomic identification could not be achieved at the species level, the ASV was identified at the lowest possible level on the phylogenetic tree.

**Figure 8 toxins-16-00102-f008:**
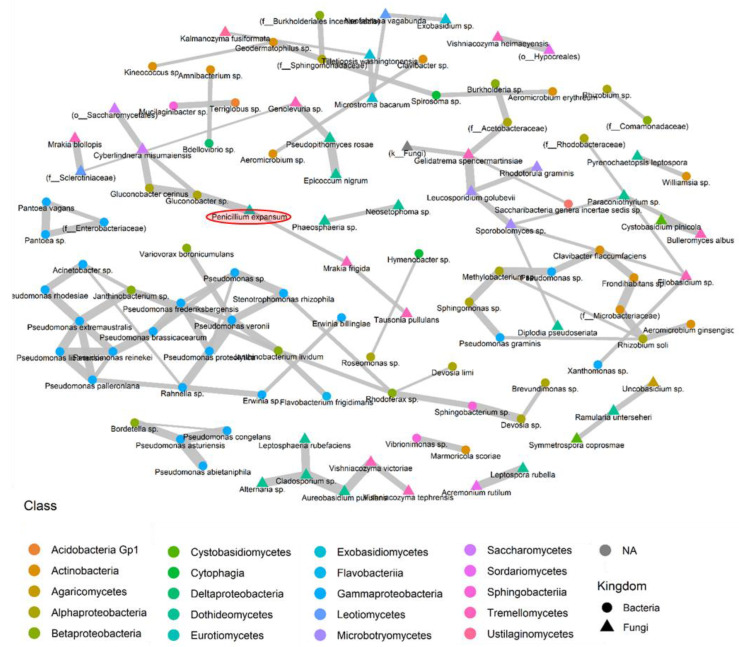
Co-occurrence network diagram, generated through SPIEC-EASI, depicting fungal and bacterial species present in a minimum of 5 samples and illustrating cooperative associations (positive interactions). Each node represents a microbe within the microbiome, and each gray link signifies pairwise co-occurrence. The thickness of the line indicates the degree of interaction between two species, with thicker lines denoting stronger interactions. In cases where taxonomic identification could not be achieved at the species level, the ASV was identified at the lowest possible level on the phylogenetic tree.

**Figure 9 toxins-16-00102-f009:**
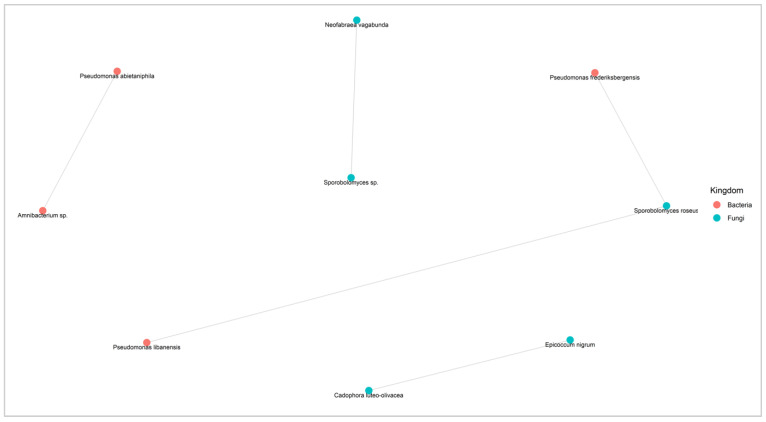
Co-occurrence network diagram (based on SPIEC-EASI) of the fungal and bacterial species present in at least 5 samples showing competitive associations (negative interactions). Each node represents a microbe from the microbiome and each gray link represents pairwise co-occurrence. When the taxonomic identification was not possible to the species level, the ASV was identified by the lowest possible level of the phylogenetic tree.

**Figure 10 toxins-16-00102-f010:**
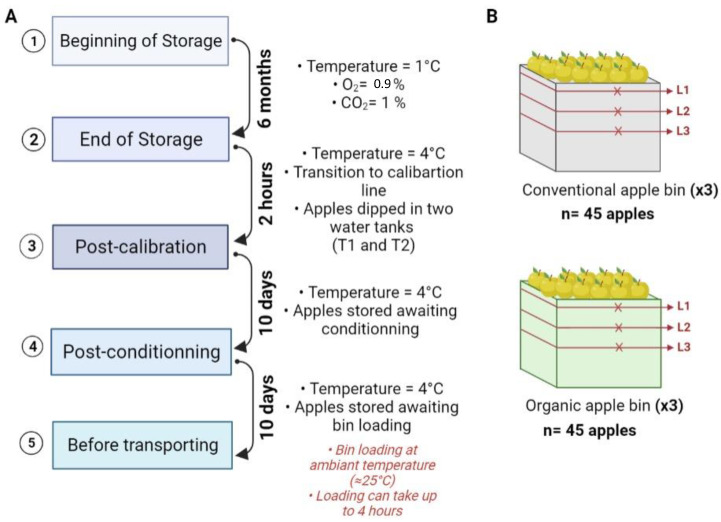
(**A**) Flow diagram of the different sampling steps, with the duration between two steps and the storage conditions of apples at each step. (**B**) Illustrations of the storage bins: conventional and organic apple bins. Three levels of sampling were determined per bin (L1, L2 and L3). Each “x” mark represents a sample composed of fives apples picked randomly at each level. After harvest, three bins per cultivation system were selected and followed up throughout the whole production chain. The number of sampled apples at each step, per cultivation system, was *n* = 45 apples. The sketches are given for illustration purposes and do not reflect the exact location of the sample.

**Figure 11 toxins-16-00102-f011:**
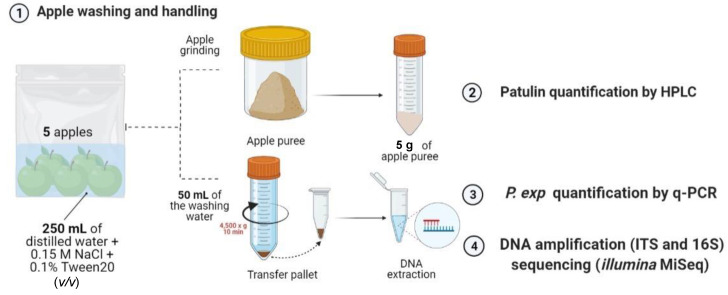
Overview of the analysis carried out on one sample (composed of 5 apples). The apples are washed with a solution of 0.15 M NaCl + 0.1% Tween 20 (*v*/*v*) in sterile conditions. The apples are then removed, ground and analyzed for their patulin content. The washing solution is centrifuged and the pellet is used for DNA extraction, followed by the quantification of *P. expansum* by q-PCR and the amplification and sequencing of the 16S and ITS regions of the epiphytic DNA of apples.

**Table 1 toxins-16-00102-t001:** Patulin concentration in apple samples obtained at various stages of the apple production chain.

Sampling Stage	Cultivation System	*n* ^1^	Number of Positive Samples ^2^ and Number of Samples Exceeding EU Limit ^3^	Patulin Content Range ^4^ (µg kg^−1^ of Apple)	Patulin Values for Positive Samples (µg kg^−1^ of Apple)
Beginning of storage	Conventional	9	2 (2)	ND—52.5	25.2; 52.5
	Organic	9	0 (0)	ND	-
End of storage	Conventional	9	2 (2)	ND—62.2	33.3; 62.2
	Organic	9	3 (3)	ND—357.7	26.5; 69.7; 357.7
Post-calibration	Conventional	9	1 (0)	ND—15.2	15.2
	Organic	9	0 (0)	ND	-
Post-conditionning	Conventional	9	0 (0)	ND	-
	Organic	9	1 (1)	ND—189.2	189.2
Before transport	Conventional	9	3 (2)	ND—147.2	15; 34.5; 147.2
	Organic	9	2 (1)	ND—46.8	12.4; 46.8

^1^ Number of analyzed samples; *n* = 9 samples per step (each sample composed of five apples). ^2^ Number of samples where patulin was detected. ^3^ EU limit for patulin in solid apple products, such as apple compote and apple puree for consumption = 25 µg kg^−1^ (EC, 2006). ^4^ Range of patulin concentrations across all samples with the lowest and highest values reported. The limit of detection (LOD) and limit of quantification (LOQ) of the methods were: 0.3 µg kg^−1^ and 1 µg kg^−1^, respectively. Instances where patulin levels were below the detection limit are labeled as “not detected” (ND).

**Table 2 toxins-16-00102-t002:** DNA quantities of *P. expansum* on the surfaces of apples harvested at various postharvest stages.

Sampling Step	Cultivation System	*n* ^1^	Number of Positive Samples ^2^	Mean (Positive Samples) ± SE ^3^ (Log Copies *P. expansum* DNA/g of Apple)	*P. expansum* Content Range (Log *P. expansum* DNA/g of Apple)
Beginning of storage	Conventional	9	5	2.6 ± 0.2	0—2.7
	Organic	9	1	0.8	0.8
End of storage	Conventional	9	4	2.8 ± 0.3	0–3.3
	Organic	9	5	4.4 ± 1.8	0–6.7
Post-calibration	Conventional	9	6	3.0 ± 1.1	0–6.0
	Organic	9	1	2.8	2.8
Post-conditionning	Conventional	9	7	4.8 ± 1.4	0–6.8
	Organic	9	6	3.8 ± 0.7	0–4.5
Before transport	Conventional	9	5	4.6 ± 1.8	0–7.7
	Organic	9	9	5.2 ± 1.3	3.0–6.2

^1^ Number of analyzed samples; *n* = 9 samples per step (each sample composed of five apples). ^2^ Number of samples where *P. expansum* was quantified. ^3^ Mean of *P. expansum* DNA amounts across positive samples, in the case of one positive sample the singular value is presented; SE: Standard Error.

## Data Availability

The data presented in this study are available on request from the corresponding authors.

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
