# Peer review of "The Influence of Long-Term Storage on the Epiphytic Microbiome of Postharvest Apples and on Penicillium expansum Occurrence and Patulin Accumulation"

_toxins, 2024, doi:10.3390/toxins16020102_

Round 1
Reviewer 1 Report
Comments and Suggestions for Authors
The paper is very well organised and written , it gives new insights on the post-harvest and long-term storage effects on the formation of fungi and bacteria including their intra-species relationships in apple. This study also provide more detailed evaluation on the effects of fungal growth of Penicillium expansum and patulin contents in apples. Considering the high risk concerns of this mycotoxin, new insights of the biocontrol of fungal growth may give very high value to this publication. It also indicates remaining issues of patulin growth even under the cold storage conditions and need for proper monitoring of potential contamination of these fruits.
The abstract is clear and covers all the parts of the paper.
As well, the brief Introduction part describes the problem and the new insights of the study.
The section dedicated to patulin content analysis is very short and presents only brief description of the table data. It is interesting that there was so high difference in concentrations of patulin in organic samples after the end of storage, which is suprisingly, considering the lower concentrations during other stages. The explanation about such levels increasing the allowed level of 50 µg.kg-1 should be explained.
It would be recommmended to discuss later the patulin concentration levels with that reported by other authors and better evaluate the coocurence of patulin contents and fungal contents.
The other sections of fungal and bacterial prevalence and correlatin analysis are well written.
There are somewhere some corrections needed. For example, µg/kg and µg.kg-1 forms are used (recommmended to use µg kg-1 (with minus 1 with Superscript).
As well, there are somewher P. expansum used in full form Penicillinum expansum and in some cases itallics is needed – lines 123, 721 ).
It is not clear from Fig 11, what are 10 mg, whereas authors mention 5 g of apple pure.
Otherwise this is very valuable paper.
Comments on the Quality of English Languageno
Reviewer 2 Report
Comments and Suggestions for Authors
Dear editor,
Thank you very much for considering me as a reviewer for the manuscript entitled “The influence of long-term storage on the epiphytic microbiome of postharvest apples and on Penicillium expansum occurrence and patulin accumulation”. The manuscript is of high quality, it is well organized and clear. This work is focused on the investigation of the effect of multiple post-harvest stages on the incidence of P. expansum and patulin contamination in apples of the variety “Golden Delicious”, conventionally or organically cultivated and destined for industrialization. Moreover, the analysis done in this study provides a basis for further fungal and bacterial interactions for the fruit disease bio-control.
Following is a list of suggestions for the authors to address to improve the manuscript:
It is not clear if there evident signs of blue mould decay were found when/where P. expansum was isolated; please clarify this in the results or discussion section.
The results of patulin in apples are expressed in ug/kg of apple, but the calibration curve is expressed in ng/ml, please convert accordingly to have it expressed in the same units.
Line 7: replace contamination for infection
Line 10: replace contaminating for infecting
Line 29: replace culture for cultivation
Line 37: update reference to last year, 5 years have passed from 2018.
Line 56-59: it is also regulated in other regions or countries, please include.
Line 473: rephrase “This step was critical for both patulin and P. expansum contamination” for “This step was critical for patulin contamination and P. expansum infection”
Line 476: replace contamination for infection
Line 596: replace produces by produced
Line 610: remove extra space
Line 623: write pullulans in italics
Line 808: replace biologically for organically to make it consistent
Reviewer 3 Report
Comments and Suggestions for Authors
This work identified two critical control points of contamination of apples by P. expansum in post-harvest stages and evaluated the risk of patulin accumulation in the apple-based products. Meanwhile, this study evaluated the effect of a long-term storage and multiple processing steps on the epiphytic microbiome of apples and the effect of the latter on the growth and toxinogenesis of P. expansum. This work could provide a basis for further fungal and bacterial interactions analysis for the fruit disease biocontrol. However, minor revision is required before publication of this article.
In the introduction, the bibliography should be oriented on the main goal of the article: the fruit microbiome.
Concerning the figures: Figure 1 is fuzzy and needs to be changed to a clearer version.
In addition, the interpretation of results has to be pushed further. The results presented in Figures or Tables do not have to be also described in the text.
Comments on the Quality of English LanguageThe manuscript need an minor editing of English language and style in order to be comprehensible.
Reviewer 4 Report
Comments and Suggestions for Authors
I think that in the conclusions there is quite a lot of information that can also be found in the discussions and as such I think that only the conclusion part should remain in a few sentences
Comments on the Quality of English LanguageMinor editing of English required
